# On Hierarchies of Fairness Notions in Cake Cutting: From Proportionality to Super Envy-Freeness

**Arnav Mehra**
Capital One
mehraarn000@gmail.com

**Alexandros Psomas**
Purdue University
apsomas@purdue.edu

## Abstract

We consider the classic cake-cutting problem of producing fair allocations for $n$ agents, in the Robertson–Webb query model. In this model, it is known that: (i) proportional allocations can be computed using $O(n \log n)$ queries, and this is optimal for deterministic protocols; (ii) envy-free allocations (a subset of proportional allocations) can be computed using $O\left(n^{n^{n^{n^{n^n}}}}\right)$ queries, and the best known lower bound is $\Omega(n^2)$; (iii) perfect allocations (a subset of envy-free allocations) cannot be computed using a bounded (in $n$) number of queries.

In this work, we introduce two hierarchies of new fairness notions: Harmonically Coalition-Resistant (HCR) and Linearly Coalition-Resistant (LCR). An allocation is HCR-$k$ if the allocation is complete and, for any subset of agents $S$ of size at most $k$, every agent $i \in S$ believes the value of all pieces allocated to agents in $S$ to be at least $\frac{1}{n-|S|+1}$, making the union of all pieces allocated to agents not in $S$ at most $\frac{n-|S|}{n-|S|+1}$; for LCR-$k$ allocations, these bounds become $\frac{|S|}{n}$ and $\frac{n-|S|}{n}$, respectively. Intuitively, these notions of fairness ask that, for every agent $i$, the collective value (from the perspective of agent $i$) that a group of agents receives is limited. If the group includes $i$, its value is lower-bounded, and if the group excludes $i$, it is upper-bounded, thus providing the agent some protection against the formation of coalitions.

Our hierarchies bridge the gap between proportionality, envy-freeness, and super envy-freeness. HCR-$k$ and LCR-$k$ coincide with proportionality for $k = 1$. For all $k \leq n$, HCR-$k$ allocations are a superset of envy-free allocations (i.e., easier to find). On the other hand, for $k \in [2, \lceil n/2 \rceil - 1]$, LCR-$k$ allocations are incomparable to envy-free allocations. For $k \geq \lceil n/2 \rceil$, LCR-$k$ allocations are a subset of envy-free allocations (i.e., harder to find), while LCR-$n$ coincides with super envy-freeness: the value of each agent for their piece is at least $1/n$, and their value for the piece allocated to any other agent is at most $1/n$.

We prove that HCR-$n$ allocations can be computed using $O(n^4)$ queries in the Robertson–Webb model. On the flip side, finding HCR-2 (and therefore all HCR-$k$ for $k \geq 2$) allocations requires $\Omega(n^2)$ queries, while LCR-2 (and therefore all LCR-$k$ for $k \geq 2$) allocations cannot be computed using a bounded (in $n$) number of queries. Our results reveal that envy-free allocations occupy a curious middle ground, between a computationally impossible notion of fairness, LCR-$\lceil n/2 \rceil$, and a computationally "easy" notion, HCR-$n$.

## 1 Introduction

We consider the classic problem of cake cutting, proposed by Hugo Steinhaus [Ste48], while in hiding during World War II [Wik25]. In this problem, there is a heterogeneous resource, the "cake,"

typically represented by the interval $[0, 1]$. There is a set $N$ of $n$ agents with different valuations functions $V_1, \ldots, V_n$ over subsets of the cake; these valuations functions are induced by probability measures over $[0, 1]$ (and hence, for example, the value of every agent for the entire cake is equal to 1). The goal in this problem is to allocate to each agent a piece of the cake — a finite union of disjoint intervals — in a fair manner, under various notions of fairness. This simple model has served as a cornerstone for fair division, shaping many of the field's foundational questions.

A major focus has been the complexity of computing fair allocations. The complexity of discrete cake-cutting protocols is measured by the number of queries they require in the query model suggested by Robertson and Webb [RW98] and later formalized (and named) by Woeginger and Sgall [WS07]. For example, it is well understood that *proportional allocations*, those where every agent $i \in N$ values their piece at least $1/n$, can be computed using $O(n \log n)$ queries [EP84]; and, this is tight: every deterministic protocol requires $\Omega(n \log n)$ queries [EP11]. Another major notion of fairness is *equitability*, which requires that every agent has the same value for the piece allocated to them, i.e., for all agents $i, j \in N$, $V_i(A_i) = V_j(A_j)$ (where $A_\ell$ is the piece allocated to agent $\ell$). Procaccia and Wang [PW17] proved that equitable allocations cannot be computed in a bounded (in $n$) number of queries. This result, in turn, rules out bounded protocols for another major notion of fairness, *perfection*: an allocation $A$ is perfect if for all agents $i, j \in N$, $V_i(A_j) = 1/n$.[1]

Arguably the most important notion of fairness — the holy grail of fair division — is *envy-freeness*; an allocation $A$ is envy-free if every agent prefers their piece to the piece allocated to any other agent, i.e., for all agents $i, j \in N$, $V_i(A_i) \geq V_i(A_j)$. It is easy to see that every perfect allocation is envy-free, and every envy-free and complete ($\cup_i A_i = [0, 1]$) allocation is proportional. Unlike proportionality, equitability, and perfection, the complexity of computing (complete) envy-free allocations has been rather elusive. The existence of *bounded* protocols remained open for two decades until Aziz and Mackenzie [AM16a] presented a protocol that requires at most $O(n^{n^{n^{n^{n^n}}}})$ queries. The best known lower bound is $\Omega(n^2)$ [Pro09], leaving an astronomical gap in our understanding.

In this paper, we zoom in on the landscape of fairness between proportionality and perfection, focusing on the query complexity of different notions in this less understood intermediate space.

## 1.1 Our contributions

We introduce two new hierarchies of fairness: Harmonically Coalition-Resistant (HCR) and Linearly Coalition-Resistant (LCR). An allocation $A$ satisfies HCR-$k$ if, for every subset of agents $S$ such that $|S| \leq k$, and for every agent $i \in S$, the value of $i$ for the union of pieces allocated to agents not in $S$ is at most $\frac{n-|S|}{n-|S|+1}$, i.e., $\sum_{j \notin S} V_i(A_j) = V_i(A_{N \setminus S}) \leq \frac{n-|S|}{n-|S|+1}$. Equivalently, $V_i(A_S) \geq \frac{1}{n-|S|+1}$. For LCR-$k$, the upper bound is $\frac{n-|S|}{n}$, and the lower bound is $\frac{|S|}{n}$. The existence of HCR-$k$ allocations is implied by the existence of envy-free allocations. Let $A$ be an envy-free allocation; for all agents $i, j \in N$, $V_i(A_j) \leq V_i(A_i)$. Adding up for all $j \notin S$ gives $\sum_{j \notin S} V_i(A_j) \leq (n - |S|)V_i(A_i) \leq (n - |S|)(1 - \sum_{j \notin S} V_i(A_j))$; re-arranging gives the desired inequality. The existence of LCR-$k$ allocations is similarly implied by the existence of perfect allocations.

Our new fairness notions are interesting precisely because they occupy a natural but largely unexplored middle ground between proportionality and envy-freeness. HCR-$k$ allocations limit how much value groups of agents collectively derive from other agents' shares; thus, they may be particularly suited for contexts where collective perceptions of fairness are important. LCR-$k$ allocations impose stronger conditions, which makes them appealing when perfection is desired (where the computational cost is prohibitive/unbounded). By providing a spectrum of fairness guarantees, these notions offer a more nuanced toolkit, enabling practitioners to better navigate the trade-off between computational complexity and fairness requirements.

In Appendix A we start by showing how our new notions relate to the existing notions of fairness; we focus on complete allocations (as, otherwise, envy-freeness is the easiest notion, since it is satisfied by an empty allocation). We show that the entire HCR-$k$ hierarchy lies between proportionality and envy-freeness, with HCR-1 being proportionality, but HCR-$n$ being a strict superset of envy-freeness. On the other hand, the entire LCR-$k$ hierarchy lies between proportionality and perfection, with LCR-1 also being proportionality, while LCR-$n$ is equivalent to *super envy-freeness*. An allocation

---

[1]Which, perhaps surprisingly, always exist [Lia40, Alo87]!

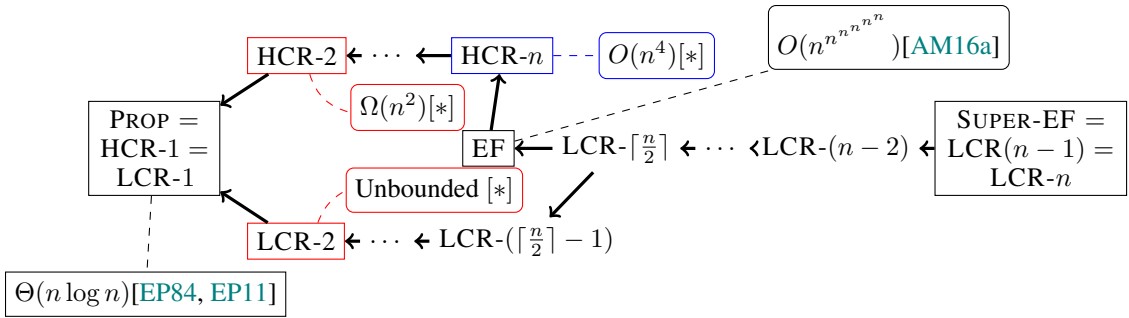

Figure 1: Relation between our new notions of fairness, Proportionality (PROP), Envy-Freeness (EF), and Super Envy-Freeness (SUPER-EF). A solid arrow from $Y$ to $X$ represents that notion $Y$ implies notion $X$, i.e. $X \leftarrow Y$ means $X \supsetneq Y$. $X \leftarrow Y$ implies that finding an allocation with property $Y$ is a harder task. Our results, highlighted in colored rectangles, are marked with $[*]$.

$A$ is super envy-free if, for all agents $i, j \in N$, $V_i(A_i) \geq 1/n \geq V_i(A_j)$. Regarding LCR-$k$ and envy-freeness, for $2 \leq k \leq \lceil \frac{n}{2} \rceil - 1$, LCR-$k$ is incomparable to envy-freeness; for $k \geq \lceil \frac{n}{2} \rceil$, LCR-$k$ implies envy-freeness. See Figure 1.

We proceed to study the complexity of computing our new notions in the Robertson-Webb model. In Section 3 we prove our main upper bound, Theorem 1: there exists a protocol that computes complete, HCR-$n$ allocations, using at most $O(n^4)$ queries in the Robertson-Webb model. The first key observation is that, a complete, proportional and "near-perfect" allocation $A = (A_1, \ldots, A_n)$ satisfies all the constraints required by HCR-$n$; concretely, completeness, along with $V_i(A_i) \geq \frac{1}{n}$ and $V_i(A_j) \geq \frac{1}{2n}$ for all $j$, implies HCR-$n$. Therefore, it is natural to start with an $\epsilon$-perfect allocation — which can be computed using only $O(n^3/\epsilon)$ queries [BM15] — and try to adjust it to satisfy the additional proportionality property. An obstacle in this approach is that, for any $\epsilon > 0$, it might be the case that the $\epsilon$-perfect allocation is such that $V_i(A_j) > \frac{1}{n}$, for all $j \neq i$, making it impossible to satisfy proportionality for $i$, unless we trim other agents' pieces. This obstacle can be bypassed with the following trick: introduce $\ell$ phantom agents, with arbitrary valuation functions, and find an $\epsilon$-perfect partition for the instance with $n + \ell$ agents. Then, the "real" agents get almost equal allocations, no one gets a piece whose value is larger than $\frac{1}{n}$, and the allocations of the phantom agents can be combined into a new piece, the residue, to be allocated among the real agents, using, e.g., a clever recursion. However, intuitively, such a step forces us to find an allocation of the residue that is weighted proportional, where agents have different weights (or, equivalently, find an allocation such that agents' utilities meet specific utility lower bounds). Unfortunately, this leads us to a dead-end: there is no bounded protocol for finding a proportional division with unequal shares [CF20]. The final piece of this puzzle is that pieces returned by the $\epsilon$-perfect call do not need to be matched to specific "real" agents, but should be available for anyone to claim.

In more detail, our algorithm, Algorithm 1, starts by finding an $\epsilon$-perfect allocation $B$ for $4n/3$ agents, where the $n/3$ extra agents, agents $n + 1, \ldots, 4n/3$, have arbitrary valuation functions. Let $B_1, \ldots, B_n$ be the first $n$ pieces of this allocations; pieces $B_{n+1}, \ldots, B_{4n/3}$ are combined into the residue $R$. Importantly, "real" agents are *not* allocated any of the $B_j$ pieces (yet). Algorithm 1 then proceeds à la Dubins-Spanier [DS61] (or Last Diminisher [Ste48]). Let $S$ be the subset of the $B_j$ pieces still available. The algorithm asks every agent $i$ to make a mark on $R$, such that their value to the left of the mark is equal to $1/n - \max_{B \in S} V_i(B)$. The agent $i^*$ with the left-most mark (breaking ties arbitrarily) gets the two corresponding pieces, one from cutting the residue and their favorite piece from $S$, and exits the process. The process is repeated until a single agent $j$ is left, who is allocated the entire remaining residue, and the unique piece left in $S$. We prove that Algorithm 1 computes a HCR-$n$ allocation using $O(n^4)$ queries. Importantly, our query complexity bound relies on the fact that the algorithm of Brânzei and Miltersen [BM15] for finding $\epsilon$-perfect allocation produces at most $O(n^2)$ intervals.

In Appendix B we prove our main lower bounds. First, we prove that computing a HCR-2 allocation requires $\Omega(n^2)$ queries (Theorem 6), while computing a LCR-2 allocation is impossible using a bounded protocol (Theorem 7). To prove our first lower bound, Theorem 6, we first formalize the

amount of knowledge an algorithm can have about an agent $i$ after $t$ queries. We use the notion of *active intervals*. Intuitively, if an interval $I$ is active for agent $i$ at step $t$, denoted by $I \in \Pi_i^t$, then $V_i(I)$ is known to the algorithm, but for all $I' \subsetneq I$, an adversary can pick $V_i(I')$ to be anything from $0$ to $V_i(I)$. That is, all these choices for $V_i(I')$ are consistent with the query responses up until time $t$. This definition was also at the core of the $\Omega(n^2)$ lower bound for envy-freeness, by Procaccia [Pro09]. Here, we argue that, if the adversary responds to queries as if the valuation functions are uniform for the entirety of an algorithm's execution, the allocation $A = (A_1, \ldots, A_n)$ that the algorithm outputs must be such that $A_i \in \Pi_i^T$ and $V_i(A_i) = \frac{1}{n}$, for all $i \in N$, where $T$ is the number of queries the algorithm made before terminating. For LCR-2 to be satisfied, $V_i(A_i) = \frac{1}{n}$ implies a condition on $V_i(A_j)$, that similarly needs to be "checked" (i.e., $I \in \Pi_i^T$, for some $I \subseteq A_j$) for all pairs of agents $i, j \in N$; the $\Omega(n^2)$ lower bound follows. Our second lower bound, Theorem 7, reduces the problem of finding an LCR-2 allocation to a known problem with unbounded query complexity: the problem of finding an exact division. In the exact division problem, we are given target values $w_1, \ldots, w_z$, and are asked to split the cake into $z$ pieces, so that every agent $i$ has value $w_\ell$ for piece $\ell$, for $\ell = 1, \ldots, z$. Exact division is known to be impossible to solve with a bounded protocol, even for the case of two valuation functions and $z = 2$ pieces with equal weights [RW98]. Theorem 7 immediately implies that the problem of finding a super-envy free allocation has unbounded query complexity in the Robertson-Webb model, which, to the best of our knowledge, was an open problem.

Finally, in Section 4, in light of the strong lower bounds for LCR-2, we consider approximations. The minimally weaker notion that we could hope to achieve is the following: (i) if $|S| = 1$, then $V_i(A_{\bar{S}}) \leq \frac{n-1}{n}$, for all $i \in S$ (i.e., the allocation is proportional), (2) if $|S| = 2$, then $V_i(A_{\bar{S}}) \leq \frac{n-2}{n}(1 + \delta)$, for all $i \in S$. We show that this notion is possible to achieve with a bounded protocol. More generally, we define $\delta$-LCR-$k$ to be the set of complete and proportional allocations, such that if $k \geq |S| \geq 2$, then $V_i(A_{\bar{S}}) \leq \frac{n-|S|}{n}(1 + \delta)$; we prove that the query complexity of computing a $\delta$-LCR-$n$ allocation is $O\left(\frac{n^6}{\epsilon} \frac{\ln(n/\epsilon)}{\ln(n)}\right)$. To prove this result, we give an algorithm for finding complete, proportional, and $\epsilon$-perfect allocations using $O\left(\frac{n^5}{\epsilon} \frac{\ln(1/\epsilon)}{\ln(n)}\right)$ queries, which might be of independent interest. The high-level blueprint of this algorithm, Algorithm 2, is similar to Algorithm 1: in phase one split the cake into approximately equal pieces and residue, and in phase two run a "cut-and-match Last Diminisher." Algorithm 2 requires much stricter conditions from the pieces and residue at the end of phase one. As opposed to Algorithm 1, phase one in Algorithm 2 is implemented via a recursion: at each iteration the residue is split into an $(n + 1)$-piece $\epsilon'$-perfect allocation. Out of these $n + 1$ pieces, one of them serves as the residue in the next iteration. The first $n$ pieces are carefully matched to previously computed pieces. This process is done $d$ times in total, where $d$ and $\epsilon'$ need to be carefully chosen so that the value of every agent for every "combined" piece (across iterations) is within the required bounds.

Our results provide significant insight into the spectrum of fairness notions between proportionality and super envy-freeness; see Figure 1. We observe that the two ways of strengthening proportionality, HCR-2 and LCR-2, lead to strikingly different lower bounds on the corresponding query complexity. Envy-freeness occupies a curious middle ground. The minimally weaker fairness notion, HCR-$n$, can be solved in $O(n^4)$ queries, while the minimally stronger fairness notion, LCR-$\lceil \frac{n}{2} \rceil$, has unbounded query complexity. A concrete take-home message of our work regarding the query complexity of finding envy-free allocations is that, if a super polynomial lower bound exists, then there must exist a subproblem strictly harder than finding HCR-$n$ allocations which requires a super polynomial number of queries to solve.

## 1.2 Related work

As already discussed, proportional allocations can be computed using $O(n \log n)$ queries [EP84], and this is tight for deterministic protocols [EP11]. Randomized algorithms can bypass this lower bound [EP06]. Surprisingly, finding a (weighted) proportional allocation when agents have unequal shares requires an unbounded number of queries [CF20].

Regarding envy-freeness, the cut-and-choose method gives an envy-free allocation for $n = 2$ agents, and the Selfridge–Conway procedure gives an envy-free allocation for $n = 3$ agents. The problem of finding an envy-free allocation for four agents, using a bounded number of queries, was resolved by [AM16b], and later improved by [ACF$^+$18]. For $n$ agents, the best known protocol requires

$O(n^{n^{n^{n^{n^{n}}}}})$ queries, while the currently best known lower bound is $\Omega(n^2)$ [Pro09]. Equitable allocations, and therefore perfect allocations, cannot be found in a bounded number of queries [PW17].

Approximate solutions are typically easier to find. Brânzei and Miltersen [BM15] prove that finding an $\epsilon$-perfect allocation requires at most $O(n^3/\epsilon)$ queries, and Brânzei and Nisan [BN22] prove that finding an $\epsilon$-envy free and connected allocation requires at most $O(n/\epsilon)$ queries (and at least $\Omega(\log(1/\epsilon))$ queries. Cechlárová and Pillárová [CP12] prove that finding an $\epsilon$-equitable and proportional allocation requires at most $O(n(\log n + \log(1/\epsilon)))$ queries.

Segal-Halevi and Suksompong [SHS20, SHS23] and Segal-Halevi and Nitzan [SHN19] study cake cutting among groups. Specifically, Segal-Halevi and Suksompong [SHS20, SHS23] study contiguous and envy-free cake cutting among groups (where agents within a group also get a contiguous piece). Segal-Halevi and Nitzan [SHN19] study a problem where a cake must be divided and allocated to (pre-determined) groups of agents, and study the existence and query complexity of various fairness notions in this model. For example, in their unanimous FS allocations the $n$ agents are placed in $k$ groups, $K$, of size $n/k$, and for any group $S \in K$ and any agent $i \in S$ it must hold that $V_i(A_S) \geq \frac{k}{n}$. This condition is quite similar to LCR-$k$; however, it is considerably less restrictive, as it places no constraints on subsets of size less than $k$, or different groupings of the agents (LCR-$k$ considers *all* possible groupings of size at most $k$). Segal-Halevi and Nitzan [SHN19] prove that unanimous-FS allocations require infinite queries; yet little is known about how close one can get with a bounded number of queries. Our algorithms provide allocations as close to LCR-$k$ as possible, with a bounded number of queries.

Berliant et al. [BTD92], and later Husseinov [Hus11], study a notion of group fairness, *group envy-freeness*, where, similar to our work, the subsets are not predefined. However, their definition only considers groups of equal size: they call an allocation $A = (A_1, \ldots, A_n)$ group envy-free, if for every pair of groups of agents $C_1$ and $C_2$, with $|C_1| = |C_2|$, there is no partition $\{B_i\}_{i \in C_1}$ of $\cup_{j \in C_2} A_j$, such that, for all $i \in C_1$, $i$ prefers $B_i$ to $A_i$, with strict preference for at least one $i \in C_1$. This notion is stronger ("harder to achieve") than perfection; the focus of the aforementioned works is existence of such allocations, and compatibility with economic efficiency.

Further afield, numerous papers study fair allocation of indivisible items among groups of agents [CFSV19, FT22, CLS25, MS22, STZ23, KSV20, AR21].

## 2   Preliminaries

We have an infinitely divisible resource, the "cake," denoted by the interval $[0, 1]$. A piece of cake refers to a finite set of disjoint intervals of $[0, 1]$. Our goal is to allocate the cake among a set $N$ of $n$ agents. An allocation $A = (A_1, \ldots, A_n)$ of the cake to the agents consists of $n$ pieces, where $A_i$ is the piece allocated to agent $i$, such that $A_i \cap A_j = \emptyset$ for all pairs of agents $i, j$. An allocation $A$ is *complete* if $\cup_{i=1}^n A_i = [0, 1]$. For an allocation $A$ and a subset of agents $S \subseteq N$, we will use notation $A_S = \cup_{i \in S} A_i$ to denote the union of all pieces allocated to the agents in $S$. We also use notation $\bar{S} = N \setminus S$ for the complement of a set $S$.

Every agent $i \in N$ has a valuation function $V_i$ that assigns a non-negative value to any subinterval of $[0, 1]$. It is convenient to think of these values as being induced by a density function $v_i$. That is, for an interval $I = [a, b]$, $V_i(I) = \int_{x=a}^b v_i(x)dx$. Valuations are (i) normalized: $V_i([0, 1]) = \int_{x=0}^1 v_i(x)dx = 1$, (ii) additive: for a set of disjoint intervals $I_1, \ldots, I_m, V_i(\cup_{j=1}^m I_j) = \sum_{j=1}^m V_i(I_j)$, (iii) non-atomic: $\forall x, y \in [0, 1], \lambda \in [0, 1], \exists z \in [x, y]$ where $V_i([x, z]) = \lambda \cdot V_i([x, y])$.

### 2.1   Robertson-Webb Model

We study the complexity of cake-cutting algorithms in the model suggested by Robertson and Webb [RW98] and later formalized (and named) by Woeginger and Sgall [WS07]. This model allows for two types of queries:

1. $\text{EVAL}_i(I)$: Given an agent $i$ and an interval $I \subseteq [0, 1]$, this query returns $V_i(I)$.

2. $\text{CUT}_i(x, v)$: Given an agent $i$, a point $x \in [0, 1]$, and a value $v \in [0, 1]$, this query returns the smallest point $x' \in [x, 1]$ such that $V_i([x, x']) = v$.

To the best of our knowledge, every discrete cake-cutting algorithm can be (and has been) analyzed in this model. As a simple example, the cut-and-choose algorithm can be implemented by two queries as follows. First, a $\text{CUT}_1(0, 1/2)$ will return the point $y$ such that $V_1([0, y]) = V_1([y, 1]) = 1/2$. The algorithm can determine whether $[0, y]$ or $[y, 1]$ should be allocated to agent 1, by making a $\text{EVAL}_2([0, y])$ and checking whether the value returned is at most $1/2$.

## 2.2 Fairness notions

Our goal is to produce allocations that are fair. An allocation $A$ is **proportional** if $V_i(A_i) \geq 1/n$ for all $i \in N$. An allocation $A$ is **envy-free** if for all $i, j \in N$, $V_i(A_i) \geq V_i(A_j)$. An allocation $A$ is $\epsilon$-**perfect** if for all $i, j \in N$, $1/n + \epsilon \geq V_i(A_j) \geq 1/n - \epsilon$; an allocation $A$ is **perfect** if it is 0-perfect. Finally, an allocation, $A$, is **super envy-free** if every agent values their own piece at least $1/n$, and values any other agent's piece at most $1/n$, i.e., $V_i(A_i) \geq \frac{1}{n} \geq V_i(A_j)$, for all $i, j \in N$.

We write PROP, EF, SUPER-EF, and PERF for the set of all complete proportional, envy-free, super envy-free, and perfect allocations, respectively. For complete allocation, every perfect allocation is super envy-free, every super envy-free allocation is envy-free, and every envy-free allocation is proportional, and there exist perfect allocations that are not super envy-free, super envy-free allocations that are not envy-free, as well as envy-free allocations that are not proportional; that is, PROP $\supsetneq$ EF $\supsetneq$ SUPER-EF $\supsetneq$ PERF.

In this paper, we define the following new notions of fairness.

**Definition 1** (Harmonically Coalition-Resistant-$k$ (HCR-$k$)). Let $k$ be an integer, such that $n \geq k \geq 1$. An allocation $A$ is Harmonically Coalition-Resistant-$k$ (HCR-$k$) if it is complete and, for every non-empty subset of agents $S \subseteq N$ such that $|S| \leq k$, and every $i \in S$, $V_i(A_{\bar{S}}) = \sum_{j \notin S} V_i(A_j) \leq \frac{n-|S|}{n-|S|+1}$.

It is easy to see that the definition of Harmonically Coalition-Resistant-1 coincides with the definition of proportionality. Slightly overloading notation, we write HCR-$k$ for the set of all complete Harmonically Coalition-Resistant-$k$ allocations.

**Definition 2** (Linearly Coalition-Resistant-$k$ (LCR-$k$)). Let $k$ be an integer, such that $n \geq k \geq 1$. An allocation $A$ is Linearly Coalition-Resistant-$k$ (LCR-$k$) if it is complete, for every non-empty subset of agents $S \subseteq N$ such that $|S| \leq k$, and every $i \in S$, $V_i(A_{\bar{S}}) = \sum_{j \notin S} V_i(A_j) \leq \frac{n-|S|}{n}$.

Slightly overloading notation, again, we write LCR-$k$ for the set of all complete Linearly Coalition-Resistant-$k$ allocations. Notice that every Linearly Coalition-Resistant-$k$ allocation is Harmonically Coalition-Resistant-$k$, since $\frac{n-|S|}{n} \leq \frac{n-|S|}{n-|S|+1}$ for all $|S| \geq 1$.

## 3 An Algorithm for HCR-$n$

In this section, we prove our main upper bound: HCR-$n$ allocations can be computed using $O(n^4)$ queries in the Robertson-Webb model.

**Theorem 1.** Algorithm 1 computes a HCR-$n$ allocation using $O(n^4)$ CUT and EVAL queries.

*Proof.* We first prove that every complete and proportional allocation $A$ such that $V_i(A_j) \geq \frac{1}{2n}$, for all agents $i, j$, satisfies HCR-$n$. Towards this, consider such an allocation $A$, and an arbitrary subset of agents $S \subseteq N$ and $i \in S$. $V_i(A_{\bar{S}}) = 1 - V_i(A_S) \leq 1 - \left( \frac{1}{n} + \frac{|S|-1}{2n} \right) = \frac{2n-|S|-1}{2n} \leq \frac{n-|S|}{n-|S|+1}$, as long as $|S| \leq n - 1$. For $|S| = n$, $V_i(A_{\bar{S}}) = 0 \leq \frac{n-|S|}{n-|S|+1}$. Therefore, $A \in$ HCR-$n$.

It remains to prove that Algorithm 1 finds an allocation with these properties (completeness, proportionality, and $V_i(A_j) \geq \frac{1}{2n}$) using $O(n^4)$ CUT and EVAL queries.

Algorithm 1 first computes an $\frac{1}{4n}$-perfect allocation $B$ for $4n/3$ agents, where the $n/3$ extra agents, agents $n + 1, \ldots, 4n/3$, have arbitrary valuation functions. Let $B_1, \ldots, B_n$ be the first $n$ pieces of this allocations; pieces $B_{n+1}, \ldots, B_{4n/3}$ are combined into the residue $R$, i.e., $R = \cup_{j=n+1}^{4n/3} B_j$. Importantly, "real" agents are *not* allocated any of the $B_j$ pieces (yet).

---

**ALGORITHM 1:** An algorithm for finding HCR-$n$ allocations

---

1 Let $V_{n+1}^+, \ldots, V_{4n/3}^+$ be $n/3$ arbitrary valuation functions.

2 $B \leftarrow \frac{1}{4n}$-PERFECT$(V_1, \ldots, V_n, V_{n+1}^+, \ldots, V_{4n/3}^+)$.     ▷ Find an $\epsilon$-PERFECT allocation with the extra agents

3 $R \leftarrow \cup_{j=n+1}^{4n/3} B_j$.           ▷ The pieces of the additional agents becomes the residue

4 $M \leftarrow \{1, \ldots, n\}$.               ▷ Initialize the set of active agents.

5 **for** $X \in \{B_1, \ldots, B_{n-1}\}$ **do**

6     Ask every agent $i \in M$ to make a mark on $R$ such that the piece to the left of the mark has value $1/n - V_i(X)$.

7     Let $i^* \in M$ be the agent with the left-most mark on $R$ (breaking ties arbitrarily).

8     Let $R_{i^*}$ be the part of $R$ to the left of $i^*$'s mark.

9     $A_{i^*} \leftarrow R_{i^*} \cup X$.                 ▷ Allocate to agent $i^*$

10     $R \leftarrow R \setminus R_{i^*}$.                  ▷ Update the residue

11     $M \leftarrow M \setminus \{i^*\}$.        ▷ Remove $i^*$'s piece from $S$ and $i^*$ from $M$

12 **end**

13 For the remaining agent $i \in M$, $A_i \leftarrow B_n \cup R$.

14 **return** $A$

---

Algorithm 1 then proceeds à la Dubins-Spanier [DS61]/Last Diminisher [Ste48]. Let $S$ be the subset of the $B_j$ pieces still available. The algorithm asks every agent $i$ to make a mark on $R$, such that their value to the left of the mark is equal to $1/n - \max_{B \in S} V_i(B)$. The agent $i^*$ with the left-most mark (breaking ties arbitrarily) gets the two corresponding pieces (one from cutting the residue and their favorite piece from $S$), and exits the process; we denote their final allocation by $A_{i^*} = A'_{i^*} \cup R_{i^*}$, where $A'_{i^*}$ is their favorite piece in $S$, and (slightly overloading notation) $R_{i^*}$ is the part of the residue allocated to them. The process is repeated until a single agent $j$ is left, who is allocated the entire remaining residue, and the unique piece left in $S$.

**Query complexity.** Using the algorithm of Brânzei and Miltersen [BM15] for finding $\epsilon$-perfect allocation, computing $(B, R)$ costs $O((4n/3)^3/(1/4n)) = O(n^4)$ queries. This protocol produces at most $O(n^2)$ intervals, and therefore, at most $O(n^3)$ EVAL queries are necessary for every agent to compute the value of every interval. When the value of each interval is known, any query across any set of these intervals, which we refer to as a *super-query*, will require at most 2 queries (see proof below). Applying this, the "Last Diminisher" steps (the while loop in Algorithm 1) will cost another $O(n^2)$ queries. Marking a piece of value $\frac{1}{n} - V_i(S_j)$ on $R$ for a single agent requires 1 super-query. So, $O(n^2)$ super-queries, or $O(n^2)$ actual queries are required for this step. Therefore, the overall complexity is $O(n^4)$.

**Claim 1.** Let $\mathcal{I}$ be a collection of $z$ noncontiguous intervals. If $V_i(I_j)$ is known for all $i \in N, I_j \in I$, then any super-query over any subset of intervals of $I$ will require at most 2 "real" queries.

*Proof.* Consider the super-query $\text{EVAL}_i([x_1, x_2])$ where $x_1 \in I$ and $x_2 \in I'$ for some intervals $I, I' \in \mathcal{I}$. If $I = I'$, $x_1$ and $x_2$ belong to the same interval, meaning only 1 actual query is required. If $I \neq I'$, then $\text{EVAL}_i([x_1, x_2]) = \text{EVAL}_i([x_1, I_{\text{right}}]) + \text{EVAL}_i([I'_{\text{left}}, x_2]) + \sum_{I \in \mathcal{I}: I \text{ "between" } I, I'} V_i(I)$. Now consider the super-query $\text{CUT}_i(v_1, x_1)$ where $x_1 \in I$ for some interval $I \in \mathcal{I}$. This requires at most 2 queries: First, perform the true query $\text{EVAL}_i([x_1, I_{\text{right}}])$. Then, using this evaluation along with those of the intervals to the right of $I$ to determine the interval $I'$ the mark $x_2$ will be placed and the value $v_2 = v_1 - V_i(x_1, I'_{\text{left}})$ that should be to the left of $x_2$ in $I'$. Finally, only one actual query is required to determine the cut $x_2$, $x_2 = \text{CUT}_i(v_2, I'_{\text{left}})$. Thus, given each agent's evaluation of each noncontiguous interval, any super-query requires at most 2 actual queries.   □

**Correctness of Algorithm 1.** All allocations are complete, by construction. First, we prove that each allocation is proportional, i.e., $V_i(A_i) \geq V_i(A'_i) \geq 1/n$, which simply means that at each iteration, all remaining agents in $M$ will be able to mark out a piece of value $\frac{1}{n}$. Consider some iteration $t$. Assuming that previous iterations were completed successfully, each agent in $M$ did not have the left-most mark in $R$, $V_i(A'_j) \leq \frac{1}{n}$ for all $i \in M, j \in N \setminus M$. Let $B_t$ be the piece arbitrarily chosen to

create an allocation in iteration $t$. $V_i(R) + V_i(B_t) = 1 - \sum_{j \in N \setminus M} V_i(A'_j) - \sum_{B_j \in S \setminus B_t} V_i(B_j) \geq 1 - \frac{t-1}{n} - \frac{n-t}{n} = \frac{1}{n}$. Thus, in this iteration, all agents will be able to place a mark on the residue. Starting with the trivial case of $t = 1$ and inductively applying this logic shows that enough of $R$ will remain for each agent remaining in $M$ to place such a mark.

Regarding the second condition, $V_i(A_j) \geq \frac{1}{2n}$ for all agents $i, j$, we have the following. By the fact that $B$ is $\frac{1}{4n}$-perfect for the instance with $4n/3$ agents: $V_i(B_\ell) \geq \frac{1}{4n/3} - \frac{1}{4n} = \frac{1}{2n}$. Observing that $V_i(A_j) = V_i(A'_j) + V_i(R_j) \geq V_i(A'_j)$, and $A'_j$ is one the $B_\ell$ pieces, completes the proof. $\qquad\square$

## 4 Algorithms for Relaxations of LCR-$n$

In this section, we study approximations of LCR-$k$, aiming to bypass the strong lower bounds of Theorem 7. Since LCR-2 is impossible, but LCR-1 (aka, proportionality) is easy to achieve, the minimally weaker notion that we could hope to achieve is the following: (i) if $|S| = 1$, then $V_i(A_{\bar{S}}) \leq \frac{n-1}{n}$, for all $i \in S$ (i.e., the allocation is proportional), (ii) if $|S| = 2$, then $V_i(A_{\bar{S}}) \leq \frac{n-2}{n}(1 + \delta)$, for all $i \in S$. More generally, we have

**Definition 3** ($\delta$-LCR-$k$). Let $k$ be an integer, such that $n \geq k \geq 1$. An allocation $A$ is $\delta$-Linearly Coalition-Resistant-$k$ (LCR-$k$) if it is proportional, and, for every non-empty subset of agents $S \subseteq N$ such that $2 \leq |S| \leq k$, and every $i \in S$, $V_i(A_{\bar{S}}) = \sum_{j \notin S} V_i(A_j) \leq \frac{n-|S|}{n}(1 + \delta)$.

We prove that $\delta$-LCR-$n$ allocations can be computed efficiently, by expanding on our approach for HCR-$n$, i.e., Algorithm 1. First, it is easy to see that a complete, $\epsilon$-perfect, and proportional allocation $A$ is $(\epsilon n)$-LCR-$n$: for all $S \subseteq N$ such that $2 \leq |S|$, and $i \in S$, $V_i(A_{\bar{S}}) \leq (n - |S|)\left(\frac{1}{n} + \epsilon\right) = \frac{n-|S|}{n}(1 + \epsilon n)$. We give an algorithm, Algorithm 2, for finding a complete, $\epsilon$-perfect, and exactly proportional allocation that requires $O\left(\frac{n^5}{\epsilon} \frac{\ln(1/\epsilon)}{\ln(n)}\right)$ queries, which might be on independent interest. To the best of our knowledge, the closest results in the literature are: (i) an algorithm that finds proportional and $\epsilon$-equitable allocations using $O(n(\log n + \log(1/\epsilon))$ queries, by Cechlárová and Pillárová [CP12], and (ii) the $O(n^3/\epsilon)$ algorithm of Brânzei and Miltersen [BM15] for finding $\epsilon$-perfect allocations. Algorithm 2 immediately implies that the query complexity of finding a $\delta$-LCR-$n$ allocation is $O\left(\frac{n^6}{\delta} \frac{\ln(1/\delta)}{\ln(n)}\right)$.

**Theorem 2.** For all $\epsilon > 0$, there exists $d \in \Theta\left(\frac{\ln(1/\epsilon)}{\ln(n)}\right)$ and $\epsilon' \in \Theta\left(\frac{\epsilon}{n^2}\right)$, such that Algorithm 2, with parameters $d$ and $\epsilon'$, computes a complete, $\epsilon$-perfect, and proportional allocation using $O\left(\frac{n^5}{\epsilon} \frac{\ln(1/\epsilon)}{\ln(n)}\right)$ CUT and EVAL queries.

Algorithm 2 differs from Algorithm 1 in its computation of $S$ and the residue $R$ prior to the final "Last Diminisher" phase (i.e. line 5 in Algorithm 1, line 11 in Algorithm 2 ). In Algorithm 1, the goal was for each piece of $S$ to have value at least $\frac{1}{2n}$, but at most $\frac{1}{n}$ for any agent. In Algorithm 2, the goal is similar, but quite harder. Each piece of $S$ must have value at least $\frac{1}{n} - \tilde{\epsilon}$ but at most $\frac{1}{n}$ for any agent, for $\tilde{\epsilon} = \frac{\epsilon}{n}$; as we show, this will suffice for $\epsilon$-perfection.

To achieve this Algorithm 2 computes $S$ and $R$ recursively. Starting with the entire cake, it computes an $(n + 1)$-piece $\epsilon'$-perfect allocation, designating a single piece of the allocation as $R$ and the remaining $n$ pieces as $S$. Then it repeats this process, treating the residue $R$ itself as an entire cake, designating a piece of the new allocation as the new residue, and combining each of the remaining $n$ pieces with a distinct piece in $S$. This process is done $d$ times in total, with the goal that the expected value of each piece of $S$ (for any agent) is spaced enough between $\frac{1}{n} - \tilde{\epsilon}$ and $\frac{1}{n}$. This allows us to choose a non-zero $\epsilon'$ that keeps the potential values of each piece in $S$ for any agent within a desired range. In short, if the number of subdivisions $d$ and the margin of error for near-perfect divisions $\epsilon'$ are chosen correctly, then $\frac{1}{n} - \tilde{\epsilon} \leq V_i(B_j) \leq \frac{1}{n}$ for all $i, j \in N$.

*Proof of Theorem 2.* In Appendix C, we show that if the parameters $\epsilon'$ and $d$ are picked correctly, Algorithm 2 outputs an allocation where $V_i(A_i) \geq \frac{1}{n}$, $V_i(A_j) \geq \frac{1}{n} - \tilde{\epsilon}$, $\forall i, j \in N$ for any $\tilde{\epsilon} > 0$. By picking $\tilde{\epsilon} = \frac{\epsilon}{n}$, our allocations are $\epsilon$-perfect (in addition to proportional and complete). To see this, notice that (i) $V_i(A_j) \geq \frac{1}{n} - \tilde{\epsilon} = \frac{1}{n} - \frac{\epsilon}{n} \geq \frac{1}{n} - \epsilon$, and (ii) $V_i(A_j) \leq 1 - (n - 1)\left(\frac{1}{n} - \tilde{\epsilon}\right) =$

---

**ALGORITHM 2:** An algorithm for finding complete, $\epsilon$-perfect, and exactly proportional allocations

---

**Input:** Parameters $\epsilon'$ and $d$.

1  Let $V^+$ be an arbitrary valuation function.
2  Let $B$ be a set of $n$ empty pieces. $R \leftarrow \{[0,1]\}$         $\triangleright$ Initialize the residue as the entire cake
3  **for** $t \in \{1, \ldots, d\}$ **do**
4      $P \leftarrow \epsilon'$-PERFECT$(V_1, \ldots, V_n, V^+)$ with respect to, and normalized on, $R$.
5      $R \leftarrow P_{n+1}$
6      **for** $i \in [1, n]$ **do**
7         $B_i \leftarrow B_i \cup P_i$         $\triangleright$ Assign the other $n$ pieces in $B$ to a piece in $S$
8      **end**
9  **end**
10  $M \leftarrow \{1, \ldots, n\}$
11  **for** $X \in \{B_1, \ldots, B_{n-1}\}$ **do**
12      Ask every agent $i \in M$ to make a mark on $R$ such that the piece to the left of the mark has value $1/n - V_i(X)$.
13      Let $i^* \in M$ be the agent with the left-most mark on $R$ (breaking ties arbitrarily).
14      Let $R_{i^*}$ be the part of $R$ to the left of $i^*$'s mark.
15      $A_{i^*} \leftarrow R_{i^*} \cup X$.         $\triangleright$ Allocate to agent $i^*$
16      $R \leftarrow R \setminus R_{i^*}$.         $\triangleright$ Update the residue
17      $M \leftarrow M \setminus \{i^*\}$.         $\triangleright$ Remove $i^*$'s piece from $S$ and $i^*$ from $M$
18  **end**
19  For the remaining agent $i \in M$, $A_i \leftarrow B_n \cup R$.
20  **return** $A$

---

$1 - (n-1)\left(\frac{1}{n} - \frac{\epsilon}{n}\right) = \frac{1}{n} + \epsilon - \frac{\epsilon}{n} \leq \frac{1}{n} + \epsilon$. In the remainder of the proof, we prove that Algorithm 2 satisfies the desired property, and has query complexity $O\left(\frac{n^4}{\tilde{\epsilon}} \cdot \frac{\ln(1/\tilde{\epsilon})}{\ln(n)}\right)$.

**Query Complexity.** We claim that the algorithm requires at most $O\left(\frac{n^4}{\tilde{\epsilon}} \cdot \frac{\ln(1/\tilde{\epsilon})}{\ln(n)}\right)$ to compute an allocation such that $V_i(A_i) \geq \frac{1}{n}$, and $V_i(A_j) \geq \frac{1}{n} - \tilde{\epsilon}$. The algorithm involves (i) computing $d$ $(n+1)$-piece $\epsilon'$-perfect allocations and (ii) performing a "cut-and-match Last Diminisher." Brânzei and Miltersen give a protocol for finding $\epsilon$-perfect allocations that uses $O\left(\frac{n^3}{\epsilon}\right)$ queries [BM15], and it is well-known that traditional "Last Diminisher" requires $O(n^2)$ queries. However, it is premature to conclude that this would require $O\left(\frac{n^3}{\epsilon'}\right) \cdot d + O(n^2) \in O\left(\frac{n^3}{\epsilon'} \cdot d\right)$ queries. Many of these "queries" are performed over a collection of noncontiguous intervals, rather than a contiguous cake, i.e., they are super-queries. To account for this, similar to our analysis of Algorithm 1, we maintain each agent's valuation of each interval, so that each super-query requires at most 2 actual queries. This is shown as Claim 1 in Algorithm 1; an identical argument works here as well.

Next, we show that if we tracked each agent's evaluation of each noncontiguous interval, then the algorithm would still require just $O\left(\frac{n^3}{\epsilon'} \cdot d\right)$ (actual) queries. Prior to computing each $\epsilon'$-perfect allocation, we must normalize each agent's valuation function over $R$. While we do not have access to valuation functions, this can be achieved by scaling the output of an EVAL super-query by $\frac{1}{V_i(R)}$, or scaling the input value of a CUT super-query by $V_i(R)$. Fortunately, since $R$ is a set of intervals whose values are assumed to be known, this requires no additional queries. Thus, computing all $d$ $\epsilon'$-perfect allocations (with $(n+1)$-pieces each) requires $O\left(\frac{n^3}{\epsilon'} \cdot d\right)$ (actual) queries. As for performing our "cut-and-match Last Diminisher", at worst, an agent will perform EVAL super-queries on every piece in $S$ and $n$ CUT super-queries on $R$. Applying our prior result again, this translates to only $O(n^2)$ actual queries in total (which is dominated by $O\left(\frac{n^3}{\epsilon'} \cdot d\right)$).

Finally, we show that tracking each agent's evaluation of each noncontiguous interval requires $O(n^3 \cdot d)$ queries (which is also dominated by $O\left(\frac{n^3}{\epsilon'} \cdot d\right)$). The $\epsilon$-perfect algorithm by Brânzei and

Miltersen [BM15] produces allocations with $O(n^2)$ cuts (thus introducing $O(n^2)$ noncontiguous intervals). To incorporate each new cut $x$ on an existing interval $I$ (thus creating intervals $I_1^+ = [I_{\text{left}}, x]$ and $I_2^+ = [x, I_{\text{right}}]$) for an agent $i$, we perform 1 query $\text{EVAL}_i(I_1^+)$ to determine the values of the new intervals created ($V_i(I_2^+)$ is deduced from $V_i(I) - V_i(I_1^+)$). Thus, introducing a cut (and by extension a noncontiguous interval) only requires $O(n)$ true queries to keep our noncontiguous interval evaluations up-to-date. To account for the scaling performed in iterations performed on the residue, no additional queries are necessary as the scaling factor can be computed using the value of the residue (which is comprised of intervals of known valuations). Since our $d$ $\epsilon'$-perfect allocations introduce $O(n^2 \cdot d)$ cuts and our "cut-and-match Last Diminisher" introduces $O(n)$ cuts (one for each of the first $n - 1$ pieces allocated), tracking the desired information requires $O(n^3 \cdot d) + O(n) \in O(n^3 \cdot d)$ queries in total.

Putting these pieces together, we can conclude that Algorithm 2 computes complete and proportional allocations, where $V_i(A_j) \geq \frac{1}{n} - \tilde{\epsilon}$ for all $i, j \in N$, using $O\left(\frac{n^3}{\epsilon'} \cdot d\right) = O\left(\frac{n^4}{\tilde{\epsilon}} \cdot \frac{\ln(1/\tilde{\epsilon})}{\ln(n)}\right)$ queries in total (and therefore a complete, proportional and $\epsilon$-perfect allocation using $O\left(\frac{n^5}{\epsilon} \cdot \frac{\ln(1/\epsilon)}{\ln(n)}\right)$ queries).

**Correctness.** From our earlier discussion on query complexity and choice of parameters $d$ and $\epsilon'$, we have that the first phase of the algorithm (lines 1 to 10) end with $n$ pieces, $B_1, \ldots, B_n$, such that $\frac{1}{n} - \tilde{\epsilon} \leq V_i(B_j) \leq \frac{1}{n}$, as well as a residue. Since the final allocation of every agent $j$ is a superset of some piece $B_\ell$, we have that $V_i(A_j) \geq V_i(B_\ell) \geq \frac{1}{n} - \tilde{\epsilon}$. Following this stage of the algorithm, we can apply the same logic as done for Algorithm 1 to conclude that, for all $i \in N$, $V_i(A_i) \geq \frac{1}{n}$ after the "cut-and-match Last Diminisher" stage. Thus, by Theorem 2 we can conclude the final allocation $A$ to be $\delta$-LCR-$n$. $\square$

## Acknowledgements

The authors would like to thank Yorgos Amanatidis, Yorgos Christodoulou, John Fearnley, and Vangelis Markakis for their input in the initial development of this project. The authors would also like to thank Ariel Procaccia for the valuable discussions and suggestions. Alexandros Psomas is supported in part by an NSF CAREER award CCF-2144208, and a research award from the Herbert Simon Family Foundation.

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

# A Relations Between Fairness Notions

In this section, we show the relation between our new fairness notions, and existing fairness notions: proportionality, envy-freeness, and perfection.

## A.1 HCR-k Relations

**Theorem 3.** PROP = HCR-1 $\supsetneq$ HCR-2 $\supsetneq \cdots \supsetneq$ HCR-$(n-1)$ = HCR-$n$ $\supsetneq$ EF.

*Proof of Theorem 3.* The first equality, PROP = HCR-1 is immediate from the definition of HCR-1: for $|S| = 1$, $1 - V_i(A_i) = V_i(A_{\bar{S}}) \leq \frac{n-1}{n} = 1 - \frac{1}{n}$. It is equally easy to see that HCR-$k$ $\supseteq$ HCR-$(k+1)$, for $1 \leq k \leq n-2$, since the conditions required to satisfy the latter are a subset of the conditions required to satisfy the former notion; Claim 2 gives an allocation $A$ such that $A \in$ HCR-$k$, but $A \notin$ HCR-$(k+1)$, proving that HCR-$k$ $\supsetneq$ HCR-$(k+1)$.

**Claim 2.** HCR-$k$ does not imply HCR-$(k+1)$ for $1 \leq k \leq n-1$.

*Proof.* Let $A \in$ HCR-$k$ be an allocation such that, for all $i \in N$, $V_i(A_i) = \frac{1}{n-k+1}$, and there exists an agent $j_i \in N \setminus \{i\}$, such that $V_i(A_j) = 1 - \frac{1}{n-k+1}$ (all other $V_i(A_w)$ are 0). It is straightforward to construct such an allocation. $A \in$ HCR-$k$, since $V_i(A_{\bar{S}}) \leq 1 - \frac{1}{n-k+1} \leq \frac{n-|S|}{n-|S|+1}$ for all $S$ such that $|S| \leq k$ and $i \in S$. However, by picking $S$ such that $i \in S$, $j_i \notin S$, and $|S| = k+1$, we have that $V_i(A_{\bar{S}}) = 1 - \frac{1}{n-k+1} = \frac{n-k}{n-k+1} > \frac{n-|S|}{n-|S|+1}$, i.e., $A \notin$ HCR-$(k+1)$. $\square$

The last equality, HCR-$(n-1)$ = HCR-$n$, again immediately follows from the definitions. Finally, Claim 3 proves that HCR-$n$ $\supsetneq$ EF.

**Claim 3.** EF $\subsetneq$ HCR-$n$.

*Proof.* To see that EF $\subseteq$ HCR-$n$, consider an allocation $A \in$ EF; we will show that $A \in$ HCR-$n$. Let $S \subseteq N$ be an arbitrary subset of agents of size $|S| \leq n$. By the definition of envy-freeness, we have that, for $i, j \in N$, $V_i(A_j) \leq V_i(A_i)$. Consider $i \in S$; adding up for all $j \notin S$ we have that

$$
\begin{aligned}
V_i(A_{\bar{S}}) &\leq (n - |S|)V_i(A_i) \\
&\leq (n - |S|)V_i(A_S) && (V_i(A_i) \leq V_i(A_S)) \\
&= (n - |S|)(1 - V_i(A_{\bar{S}})).
\end{aligned}
$$

Re-arranging we get the desired inequality: $V_i(A_{\bar{S}}) \leq \frac{n-|S|}{n-|S|+1}$.

To see that HCR-$n$ does not imply EF, consider an allocation $A \in$ HCR-$n$ such that, for all agents $i \in N$ $V_i(A_i) = \frac{1}{3}$; furthermore, for every agent $i \in N$, there exist distinct agents $j_i$ and $\ell_i$ such that $V_i(A_{j_i}) = \frac{1}{2}$ and $V_i(A_{\ell_i}) = \frac{1}{6}$ (and $V_i(A_w) = 0$ for all other agents $w$). It is easy to see that such an allocation can be constructed. $A \in$ HCR-$n$, since: (i) for all $S \subseteq N$, $|S| \leq n-2$, and all $i \in S$, $V_i(A_{\bar{S}}) \leq 1 - V_i(A_i) = \frac{2}{3} \leq \frac{n-|S|}{n-|S|+1}$, (ii) for all $S \subseteq N$, $|S| = n-1$, and all $i \in S$, $V_i(A_{\bar{S}}) \leq 1 - (V_i(A_i) + \min\{V_i(A_{j_i}), V_i(A_{\ell_i})\}) \leq \frac{1}{2} \leq \frac{n-|S|}{n-|S|+1}$, and (iii) $V_i(A_{\bar{S}}) = 0$ for $S = N$. However, $A \notin$ EF, since every agent $i \in N$ envies agent $j_i$. $\square$

This concludes the proof of Theorem 3. $\square$

## A.2 LCR-k Relations

**Theorem 4.** PROP = LCR-1 $\supsetneq$ LCR-2 $\supsetneq \cdots \supsetneq$ LCR-$(n-1)$ = LCR-$n$ = SUPER-EF.

*Proof of Theorem 4.* By the definition of LCR-$k$, we have that PROP = LCR-1, LCR-$k$ $\supseteq$ LCR-$(k+1)$ for $1 \leq k \leq n-1$, and LCR-$(n-1)$ is equivalent to LCR-$n$. The theorem follows from Claim 4 and Claim 5.

**Claim 4.** LCR-$k$ does not imply LCR-$(k+1)$, for any $1 \leq k < n-1$.

*Proof.* Consider an allocation $A \in$ LCR-$k$ such that, for all $i \in N$: (i) $V_i(A_i) = \frac{k}{n}$, (ii) $V_i(A_j) = 0$, for all $j \in Z_i$, for some $Z_i \subset N \setminus \{i\}$ such that $|Z_i| = k$, and (iii) $V_i(A_j) = \left(1 + \frac{1}{n-k-1}\right) \cdot \frac{1}{n}$, for all $j \in N \setminus (Z_i \cup \{i\})$. $S_i = Z_i \cup \{i\}$; $|S_i| = k + 1$. Then, $V_i(A_{S_i}) = V_i(A_i) = \frac{k}{n}$, and therefore $V_i(A_{\bar{S}_i}) = \frac{n-k}{n} < \frac{n-|S_i|}{n}$, i.e. $A \notin$ LCR-$(k + 1)$. $\square$

**Claim 5.** LCR-$n$ = SUPER-EF.

*Proof.* Consider an allocation $A \in$ LCR-$n$. For all $S \subseteq N$ such that $|S| = n - 1$, and all $i \in S$, $V_i(A_{\bar{S}}) \leq \frac{1}{n}$; however, $\bar{S}$ has a single (arbitrary) agent. That is, $V_i(A_{\bar{S}}) = V_i(A_j) \leq \frac{1}{n}$, for all $j \in N \setminus \{i\}$. And, since LCR-$n \subseteq$ PROP, we also have $V_i(A_i) \geq \frac{1}{n}$, therefore, $A \in$ SUPER-EF.

Consider an allocation $A \in$ SUPER-EF. We have that, for all $i, j \in N$, $j \neq i$, $\frac{1}{n} \geq V_i(A_j)$. Adding up this for all $j \notin S$, for some arbitrary set $S \subseteq N$, $|S| \leq n$, such that $i \in S$, we have $V_i(A_{\bar{S}}) \leq \frac{n-|S|}{n}$, i.e., $A \in$ LCR-$n$. $\square$

This concludes the proof of Theorem 4 $\square$

**Theorem 5.** EF $\supsetneq$ LCR-$\lceil \frac{n}{2} \rceil$, but EF $\not\supseteq$ LCR-$(\lceil \frac{n}{2} \rceil - 1)$ and EF $\not\subseteq$ LCR-2.

*Proof of Theorem 5.* Consider an allocation $A \in$ LCR-$\lceil \frac{n}{2} \rceil$. Let $i, j \in N$ be two arbitrary agents. Let $S \subseteq N \setminus \{j\}$, such that (i) $i \in S$, and (ii), $|S| = \lceil \frac{n}{2} \rceil$. We have that $V_i(A_S) = 1 - V_i(A_{\bar{S}}) \geq 1 - \frac{n-|S|}{n} = \frac{1}{2} + \frac{n \mod 2}{2n}$. Let $S'$ be such that $\bar{S}' = (S \setminus \{i\}) \cup \{j\}$ (i.e., $S' = \{i\} \cup (N \setminus (\{j\} \cup S))$). We have that $|S'| = n - \lceil \frac{n}{2} \rceil = \lfloor \frac{n}{2} \rfloor$, therefore, since $A \in$ LCR-$\lceil \frac{n}{2} \rceil$, and $i \in S'$, $V_i(A_{\bar{S}'}) \leq \frac{n-|S'|}{n} = \frac{1}{2} + \frac{n \mod 2}{2n}$. Therefore, $V_i(A_S) \geq V_i(A_{\bar{S}'})$. However, $V_i(A_S) = V_i(A_i) + \sum_{z \in S \setminus \{i\}} V_i(A_z)$, and $V_i(A_{\bar{S}'}) = V_i(A_j) + \sum_{z \in S \setminus \{i\}} V_i(A_z)$. Therefore, we have that $V_i(A_i) \geq V_i(A_j)$, i.e., $i$ does not envy $j$.

Next, notice that LCR-$(\lceil \frac{n}{2} \rceil - 1)$ (and by extension, LCR-$k$ for $k < \lceil \frac{n}{2} \rceil - 1$) does not imply EF. Consider an allocation $A \in$ LCR-$(\lceil \frac{n}{2} \rceil - 1)$ such that, for all $i \in N$, $V_i(A_i) = (\lceil \frac{n}{2} \rceil - 1) \cdot \frac{1}{n}$, and there exists an agent $j_i$ such that $V_i(A_{j_i}) = 1 - (\lceil \frac{n}{2} \rceil - 1) \cdot \frac{1}{n}$, and $V_i(A_z) = 0$, for all agents $z \neq i, j_i$. Since $V_i(A_i) < V_i(A_{j_i})$, envy-freeness is violated.

Finally, EF does not imply LCR-2 (and by extension, LCR-$k$ for $k > 2$). Consider an allocation $A \in$ EF such that, for all $i \in N$ there exists an agent $j_i$, such that $V_i(A_{j_i}) = 0$, and $V_i(A_z) = \frac{1}{n-1}$, for all $z \neq i, j_i$. For $S = \{i, j_i\}$ and $n > 2$, $V_i(A_{\bar{S}}) = \frac{n-2}{n-1} > \frac{n-|S|}{n} = \frac{2}{n}$. $\square$

## B  Lower Bounds

In this section, we state our lower bounds for HCR-2 (Theorem 6) and LCR-2 (Theorem 7).

**Theorem 6.** Computing HCR-2 allocations requires $\Omega(n^2)$ queries.

*Proof.* For this proof, we use many definitions and lemmas from the work of Procaccia [Pro09], who proves the $\Omega(n^2)$ lower bound for finding envy-free allocations.

Consider an arbitrary algorithm. First, to analyze the information available to the algorithm at each step, we define, for every agent $i \in N$ and every step $t$, a set of disjoint intervals $\Pi_i^t$, that is a partition of $[0, 1]$. We say that interval $I \in \Pi_i^t$ is active with respect to agent $i$ at step $t$.

$\Pi_i^t$s are defined recursively. $\Pi_i^0 = \{[0, 1]\}$, since the only information available to the algorithm after 0 steps/queries is that $V_i([0, 1]) = 1$. Assuming that at step $t$ we have $\Pi_i^t$, if at step $t + 1$ the algorithm does not make a query for agent $i$ (i.e., the query at step $t + 1$ is EVAL$_j$ or CUT$_j$ for some $j \neq i$), then $\Pi_i^{t+1} = \Pi_i^t$. Otherwise, $\Pi_i^{t+1}$ gets updated accordingly. For example, if the query is EVAL$_i(x_1, x_2)$, where $x_1 \in I_1$ and $x_2 \in I_2$, for some intervals $I_1, I_2 \in \Pi_i^t$, then, informally,

$$\Pi_i^{t+1} = (\Pi_i^t \setminus \{I_1, I_2\}) \cup \{[left(I_1), x_1], [x_1, right(I_1)], [left(I_2), x_2], [x_2, right(I_2)]\},$$

where for an interval $I = [a, b]$, $left(I) = a$ and $right(I) = b$. Intuitively, the algorithm at step $t$ "knew" $i$'s value for $I_1$ and $I_2$, and after the EVAL$_i(x_1, x_2)$, it can infer (at most) the value of agent $i$ for

four additional intervals: $[left(I_1), x_1], [x_1, right(I_1)], [left(I_2), x_2]$, and $[x_2, right(I_2)]$ (noting, that the first two imply the value for $I_1$ and the second two imply the value for $I_2$). Procaccia [Pro09] proves two crucial lemmas:

**Lemma 1** ([Pro09]; Lemma 3.2). *For all $i \in N$ and stage $t$, $|\Pi_i^{t+1}| - |\Pi_i^t| \leq 2$.*

**Lemma 2** ([Pro09]; Lemma 3.3). *For all $i \in N$ and stage $t$, $\Pi_i^t$ has the following properties:*

1. *For every $I \in \Pi_i^t$, $V_i(I)$ is known to the algorithm at stage $t$.*

2. *For every $I \in \Pi_i^t$, $I' \subsetneq I$, and $0 \leq \lambda \leq 1$, it might be the case (based on the information available to the algorithm at stage $t$) that $V_i(I') = \lambda V_i(I)$.*

Consider an adversary that, for every agent $i \in N$, responds as if the valuation of the agent was uniform over the interval $[0, 1]$ (i.e., responds $\text{EVAL}_i([x_1, x_2]) = x_2 - x_1$, and $\text{CUT}_i(x, v) = x + v$). Let $T$ be the number of queries our algorithm asks before it terminates and outputs allocation $A = (A_1, \dots, A_n) \in \text{HCR-2}$.

First, we claim that for all $i \in N$, there exists an active interval $I_i \subseteq A_i$, such that (i) $I_i \in \Pi_i^T$, and (ii) $V_i(I_i) \geq \frac{1}{n}$. If this is not the case, one can define $V_i$ (consistently with $\Pi_i^T$) such that proportionality is violated. Concretely, (1) if $I \notin \Pi_i^T$ for all $I \subseteq A_i$ then we can concentrate the value of all $I$ such that $I \cap A_i \neq \emptyset$ to the sub-intervals outside of $A_i$ (and therefore, $V_i(A_i) = 0$), and (2) if $V_i(I) < \frac{1}{n}$ for all $I \subseteq A_i$, $I \in \Pi_i^T$, then we can pick $V_i$ such that $V_i(A_i) = \max_{I \in \Pi_i^T} V_i(I) < 1/n$.

Since all queries until time $T$ have been answered as if the valuations were uniform, it must be that $1/n \leq V_i(I_i) = |I_i|$ for all $i \in N$. However, if for all $i \in N$ we have: (i) $I_i \subseteq A_i$, (ii) $|I_i| \geq 1/n$, (iii) $A_i \cap A_j = \emptyset$ for all $j \neq i$, and (iv) $\cup_{i=1}^n A_i = [0, 1]$, then it must be that $|I_i| = |A_i| = 1/n$, as well as $V_i(A_i) = 1/n$, for all $i \in N$. The HCR-2 property then implies that $V_i(A_{N\setminus\{i,j\}}) = 1 - V_i(A_i) - V_i(A_j) \leq 1 - \frac{1}{n-1}$, or $V_i(A_j) \geq \frac{1}{n-1} - \frac{1}{n} = \frac{1}{n(n-1)}$ for all $i, j \in N$.

To meet this condition, it must be that for every agent $i$ and piece $A_j$, there exists an active interval $I \in \Pi_i^T$ such that (i) $I \subseteq A_j$, and (ii) $V_i(I) \neq 0$. Similarly to our earlier argument, if this is not the case, one can define $V_i$ (consistently with $\Pi_i^T$) such that $V_i(A_j) = 0$ $(< \frac{1}{n(n-1)})$. Concretely, we can pick $V_i$ such that $V_i(I \cap A_j) = 0$ and $V_i(I \setminus A_j) = V_i(I)$ for all $I \in \Pi_i^T$ and $I \cap A_j \neq \emptyset$.

Since allocations are pair-wise disjoint, we must then have at least $n$ active intervals per agent by time $T$, i.e. $|\Pi_i^T| \geq n$ for every agent $i$. Since $|\Pi_i^0| = 1$, Lemma 1 implies that $|\Pi_i^T| \leq 2T + 1$, and therefore $2T + 1 \geq n$, i.e. the algorithm makes at least $\frac{n-1}{2}$ queries to every agent $i \in N$. Overall, the algorithm makes at least $n \cdot \frac{n-1}{2} \in \Omega(n^2)$ queries overall. $\qquad\square$

**Theorem 7.** *Computing LCR-2 allocations requires an infinite number of queries, for all $n \geq 3$.*

*Proof.* We prove the statement for all odd $n$, $n \geq 3$; the proof can be easily adjusted to even $n$.

Consider an instance where $\lfloor \frac{n}{2} \rfloor$ agents — agents 1 through $\lfloor \frac{n}{2} \rfloor$ — have the same valuation function $V_1$, $\lfloor \frac{n}{2} \rfloor$ agents — agents $\lfloor \frac{n}{2} \rfloor + 1$ through $n - 1$ — have the same valuation function $V_2$, and the last agent, agent $n$, has valuation function $V_n(x) = \frac{V_1(x) + V_2(x)}{2}$ for all $x \subseteq [0, 1]$.

Let $A$ be a LCR-2 allocation. For every agent $i = 1, \dots, \lfloor \frac{n}{2} \rfloor$, let $j_i = i + \lfloor \frac{n}{2} \rfloor$. Since agent $i$ has valuation $V_1$, we have that $V_1([0, 1] \setminus (A_i \cup A_{j_i})) \leq 1 - 2/n$, or simply, $V_1(A_i \cup A_{j_i}) \geq 2/n$, from the definition of LCR-2. It also holds that $V_2(A_i \cup A_{j_i}) \geq 2/n$. We have $\cup_{i=1}^{\lfloor \frac{n}{2} \rfloor} (A_i \cup A_{j_i}) = [0, 1] \setminus A_n$, and $V_1([0, 1]) = V_2([0, 1]) = 1$, therefore, $V_1(A_n) = 1 - \sum_{i=1}^{\lfloor \frac{n}{2} \rfloor} V_1(A_i \cup A_{j_i}) \leq 1 - \lfloor \frac{n}{2} \rfloor \frac{2}{n} = 1 - \frac{n-1}{2} \frac{2}{n} = \frac{1}{n}$. Similarly, $V_2(A_n) \leq \frac{1}{n}$. Since LCR-2 implies proportionality, $V_n(A_n) = \frac{V_1(A_n) + V_2(A_n)}{2} \geq \frac{1}{n}$, i.e., $V_1(A_n) + V_2(A_n) \geq \frac{2}{n}$. Therefore, it must be that $V_1(A_n) = V_2(A_n) = \frac{1}{n}$.

Since $V_1(A_i \cup A_{j_i}) \geq \frac{2}{n}$, for all $i = 1, \dots, \lfloor \frac{n}{2} \rfloor$, and $\lfloor \frac{n}{2} \rfloor \frac{2}{n} = \frac{n-1}{2} \frac{2}{n} = 1 - \frac{1}{n}$ (which is exactly the value of $[0, 1] \setminus A_n$), it must be that $V_1(A_i \cup A_{j_i}) = \frac{2}{n}$. Similarly, it must be that $V_2(A_i \cup A_{j_i}) = \frac{2}{n}$.

Therefore, we overall have that, there are $\lfloor \frac{n}{2} \rfloor + 1$ pieces, $A_n$ and the $(A_i \cup A_{j_i})$s, such that, $V_1$ and $V_2$ have the same value of $\frac{1}{n}$ for the first piece, and the same value of $\frac{2}{n}$ for all remaining pieces.

Given $n$ valuations functions $U_1, \ldots, U_n$ and weights $w_1, \ldots, w_k$, asking for a partition of the cake into $k$ pieces $I_1, \ldots, I_k$ such that $U_i(I_j) = w_j$, is known as the *exact division*, or consensus splitting, problem. This problem is known to be impossible to solve with a bounded protocol, even for the case of two valuation functions and $k = 2$ pieces with equal weights [RW98]. Since finding a LCR-2 allocation would give a solution to the exact division problem for two valuation functions and $\lfloor \frac{n}{2} \rfloor + 1$ pieces, we can conclude that there is no bounded protocol for finding a LCR-2 allocation.

Note that for even $n$, the proof is near-identical: the only difference is that the $n$-th agent is not necessary (i.e., half the agents have valuation $V_1$ and the other half have valuation $V_2$), and using the same arguments we would conclude that $V_1(A_i \cup A_{j_i}) = V_2(A_i \cup A_{j_i}) = \frac{2}{n}$, for all $i = 1, \ldots, n/2$. $\qquad\square$

## C  Picking the parameters of Algorithm 2

Consider the partition, $B$, created in the $t^{th}$ iteration of line 4. Due to the normalization of $R$ at each iteration, $B$ will have the following property:

$$\left( \frac{1}{n+1} - \epsilon' \right)^t \leq V_i(B_j) \leq \left( \frac{1}{n+1} + \epsilon' \right)^t, \ \forall i, j \in N$$

Let $S'$ describe the state of $S$ prior to the final "Last Diminisher" phase of the algorithm (i.e. line 11). $S_i'$ will be comprised of 1 piece from each $\epsilon'$-perfect partition, $B$, created. As discussed, each piece of $S'$ needs to have value at least $\frac{1}{n} - \tilde{\epsilon}$ but at most $\frac{1}{n}$ for any agent.

$$V_i(S_j') \leq \sum_{t=1}^{d} \left( \frac{1}{n+1} + \epsilon' \right)^t \leq \frac{1}{n}$$

$$V_i(S_j') \geq \sum_{t=1}^{d} \left( \frac{1}{n+1} - \epsilon' \right)^t \geq \frac{1}{n} - \tilde{\epsilon}$$

Notice that by polynomial expansion, we can see that:

$$\left( \frac{1}{n+1} + \epsilon' \right)^t - \left( \frac{1}{n+1} \right)^t \geq \left( \frac{1}{n+1} \right)^t - \left( \frac{1}{n+1} - \epsilon' \right)^t$$

Therefore, if we define $d$ such that $\sum_{t=1}^{d} \left( \frac{1}{n+1} \right)^t \geq \frac{1}{n} - \frac{\tilde{\epsilon}}{2}$ (the average of the required upper and lower bounds of $V_i(S_j')$), then for increasing $\epsilon'$, the computed upper bound of $V_i(S_j')$ will exceed $\frac{1}{n}$ before the computed lower bound dips below $\frac{1}{n} - \tilde{\epsilon}$. With this in mind, we define $d$ as described:

$$\sum_{t=1}^{d} \left( \frac{1}{n+1} \right)^t = \frac{1 - \left( \frac{1}{n+1} \right)^d}{n} \geq \frac{1}{n} - \frac{\tilde{\epsilon}}{2}$$

$$d = \left\lceil \frac{\ln \left( \frac{\tilde{\epsilon} n}{2} \right)}{\ln \left( \frac{1}{n+1} \right)} \right\rceil = \left\lceil \frac{\ln(2) + \ln(1/\tilde{\epsilon}) + \ln(1/n)}{\ln(n+1)} \right\rceil \in \Theta \left( \frac{\ln(1/\tilde{\epsilon})}{\ln(n)} \right).$$

Now, we define $\epsilon'$ such that $V_i(S'_j) \leq \frac{1}{n}$, noting that if the required upper bound holds, then so will the required lower bound. For simplicity, we first loosen the upper bound:

$$V_i(S'_j) \leq \sum_{t=1}^{d} \left( \frac{1}{n+1} + \epsilon' \right)^t$$

$$= \left( \frac{1}{n+1} + \epsilon' \right) \cdot \frac{1 - \left( \frac{1}{n+1} + \epsilon' \right)^d}{1 - \left( \frac{1}{n+1} + \epsilon' \right)}$$

$$\leq \left( \frac{1}{n+1} + \epsilon' \right) \cdot \frac{1 - \left( \frac{1}{n+1} \right)^d}{1 - \left( \frac{1}{n+1} + \epsilon' \right)}$$

$$\leq \left( \frac{1}{n+1} + \epsilon' \right) \cdot \frac{1 - \frac{1}{n+1} \cdot \frac{\tilde{\epsilon}n}{2}}{1 - \left( \frac{1}{n+1} + \epsilon' \right)}.$$

Enforcing that this last expression is at most $1/n$ we have $\frac{\left( \frac{1}{n+1} + \epsilon' \right)}{1 - \left( \frac{1}{n+1} + \epsilon' \right)} \leq \frac{1}{n\left( 1 - \frac{1}{n+1} \cdot \frac{\tilde{\epsilon}n}{2} \right)}$, or

$$\epsilon' \leq \frac{1}{n \left( 1 - \frac{1}{n+1} \cdot \frac{\tilde{\epsilon}n}{2} \right) + 1} - \frac{1}{n+1} = \frac{\tilde{\epsilon}n^2}{2(n+1)^2 \left( n + 1 - \frac{\tilde{\epsilon}n^2}{2(n+1)} \right)}.$$

Considering that $\frac{\tilde{\epsilon}n^2}{2(n+1)} > 0$, we can tighten this constraint by decreasing the upper bound as so:

$$\epsilon' \leq \frac{\tilde{\epsilon}n^2}{2(n+1)^3} \in \Theta \left( \frac{\tilde{\epsilon}}{n} \right).$$

Thus, the aforementioned conditions will hold for some $d \in \Theta \left( \frac{\ln(1/\tilde{\epsilon})}{\ln(n)} \right)$ and $\epsilon' \in \Theta \left( \frac{\tilde{\epsilon}}{n} \right)$.

