# OpenReview forum: "On Hierarchies of Fairness Notions in Cake Cutting: From Proportionality to Super Envy-Freeness"
_NeurIPS.cc/2025/Conference — NeurIPS 2025 poster_

### Official Review · Reviewer_JByF · 2025-06-20

**Clarity:** 3
**Significance:** 2
**Originality:** 3
**Rating:** 3
**Confidence:** 3

**Summary:**

The paper studies the cake cutting problem. It introduces two new families of fairness notions, which both conceptually lie in between the known fairness notions of proportionality and envy-freeness. The paper gives upper and lower bounds on the query complexity of finding outcomes satisfying these fairness notions and relate them to previously defined ones. Most importantly, the paper gives an algorithm which finds a CHB-n outcome (which is one of the defined fairness notions) with O(n^4) queries.

**Questions:**

I am very much willing to increase my score, given a better intuitive motivation of (in particular) CHB and the accompanying algorithm answering (a subset of) the following questions:

What is your motivation behind CHB? Why is the CHB family of fairness notions natural and worth achieving over just proportionality? Why does an agent (implicitly) care about what other agents receive?

I might also be misunderstanding something, so I am quite open-minded regarding the rebuttal :)

**Ethical Concerns:**

["NO or VERY MINOR ethics concerns only"]

**Final Justification:**

I thank the authors for their rebuttal. I think fundamentally, we agree on the issues at hand, but perhaps we disagree at the "weights" assigned to these issues. I still feel that the motivation for the provided fairness axiom is very lacking and I am not sure the algorithmic contribution is enough for we to outweigh this (especially for NeurIPS compared to say Wine/EC/Soda, as another reviewer mentioned).  Copying from what I mentioned in the discussion: "To re-iterate, I especially find the CHB notion to be quite not so well motivated. Yes, it implies proportionality, but proportionality on its own does not (at least to me) imply fairness. Proportionality (again at least to me) seems more like a minimal "test" of fairness, so positive and negative results for proportionality are in some sense akin to mapping the boundary of possible fair algorithms. CHB much less seems like a minimal "test", but also seems far to weak to be a sufficient condition to be fair. (I hope this makes sense)".

**Limitations:**

/

**Paper Formatting Concerns:**

/

**Quality:**

3

**Strengths And Weaknesses:**

In general, cake cutting (or fair division in total) is a suitable topic for NeurIPS. The paper is well-written and for the most part easy to follow, the introduced fairness notions are interesting (in particular, since they manage to slot in somewhere between the almost "untouchable" envy-freeness and the easy-to-achieve proportionality) and I can see this paper sparking some follow-up in the fair division community.

Yet, there is something I am struggling with in this paper and which makes me question its suitability for NeurIPS.
In essence, I do not really find the introduced fairness notions to be entirely convincing and I do not think the authors really motivate them well-enough, besides the fact that they are (let's say) curious fairness notions, that lie somewhere in-between the standard ones of envy-freeness and proportionality. In particular, I am missing some motivation and discussion on why one might be interested in the utility of bundles assigned to other agents. Why do I as an agent do not just care about my own bundle and maybe the bundle of other agents in comparison to mine, but also about the total welfare (according to my own subjective opinion) of every subgroup I could possibly be a part of? Similarly, I am missing some motivation for the actual form the fairness notions take. CLB is easy to understand, but what is the motivation behind CHB (except that it works out with the algorithm and that envy-freeness implies it). I think especially for a conference like NeurIPS (which is not just purely a theory conference) a discussion on things like this is important. To add on to this, I think these prior points also lead to "weirdly-behaving" algorithms. Consider for instance the case of two agents, where the first agent uniformly values the interval [0,1/2] and does not value the rest, while the second agent only values [1/2, 1]. Here, Algorithm 1, (due to its requirement of having every agent value every other agents bundle positively) gives agent 1 some part of [1/2, 1] and vice versa for agent 2, now this is not forced by CHB, but still due to the insistence on giving every subset of agents something (in the eyes of every agent) it seems somewhat inevitable that something like this would happen in a simple algorithm satisfying CHB. Still, this makes the algorithm seem more like a theoretical novelty, and not like a "rule" or an actual algorithm which you would want to use for an actual cake cutting instance --- which in my mind would greatly improve its suitability for NeurIPS.

Minor comments:
- I am not a big fan of citing Wikipedia. For this particular use case there are better/more scientific sources on Steinhaus
- Line 169: Missing a space after "queries"
- Line 263: Might be worth making a brief comment why this is non-negative
- Line 269: should be "for finding *an* epsilon-perfect" (also I prefer the $\varepsilon$ over $\epsilon$, but this is personal preference)
- I am personally missing a discussion section in this paper
- Line 824: Sometimes you write V_i(\overline{S'}) instead of V_i(A_\overline{S'})

---

> ### Author Rebuttal · Authors · 2025-07-28
>
> Thank you for your thorough and thoughtful review.
>
> Regarding the motivation (and intuition) for CHB: one can think of proportionality as a guarantee to every agent $i$ that, from $i$'s perspective, "the remaining $n-1$ agents, combined, don’t have too much of the cake" (as opposed to the standard phrasing/viewpoint: "agent $i$ has enough of the cake"). Envy-freeness sits at the other extreme: the guarantee to every agent $i$ is that, from $i$'s perspective, "for every other agent $j$, agent $j$ doesn't have too much of the cake." CHB interpolates between the two extremes: the guarantee to every agent $i$ is that, from $i$'s perspective, "for every group of agents, the group, combined, doesn’t have too much of the cake." Put differently, proportionality limits the bundle of "everyone else combined," envy-freeness limits the bundle of "every other individual," and CLB/CHB limit the bundle of "every group." Yet another intuition for CLB/CHB is that each agent wants the top $x$% of other agents not to have more than $y$% of the cake. (Also, to answer your last question, note that all three notions impose limits on what others receive.)
>
> Viewed through this lens, we believe CHB is a natural and very well-motivated notion --- especially given our limited understanding of the complexity of envy-freeness, despite substantial efforts from the fair division community. Our results imply that CHB also offers an algorithmically tractable alternative to envy-freeness.
>
> Regarding relevance to NeurIPS, the NeurIPS community has been increasingly interested in formal models of fairness. Cake cutting is the canonical model for studying resource allocation with fairness guarantees to individual agents (as opposed to fairness guarantees to groups of agents). Therefore, we believe our work will resonate with the NeurIPS audience.
>
> The "weird behavior" of Algorithm 1 you point out is interesting; however, one can raise similar concerns for perfect allocations ($v_i(X_j) = 1/n$), which are considered extremely fair. We understand your concern as, at least implicitly, suggesting adding some form of efficiency; we agree this is an interesting and important direction. That being said, it is well known that Pareto efficiency cannot be achieved by a finite protocol in the RW model (see "How to Cut a Cake Before the Party Ends" by Kurokawa et al.). One can, of course, turn to approximations to fairness/efficiency (like "Optimal Envy-Free Cake Cutting" by Cohler et al.) or change the model a bit (like "Beyond Cake Cutting: Allocating Homogeneous Divisible Goods" by Caragiannis et al.). We find these directions compelling, but beyond the scope of this work, and leave them as promising future directions.
>
> We would be happy to incorporate these clarifications in the final version of the paper. We will also address the minor comments you point out (Wikipedia citation, typos, etc) in the final version of the paper.

---

> > ### Comment · Reviewer_JByF · 2025-08-04
> >
> > Thank you for the rebuttal.
> >
> > > CHB interpolates between the two extremes: the guarantee to every agent $i$ is that, from $i$'s perspective, "for every group of agents, the group, combined, doesn’t have too much of the cake." Put differently, proportionality limits the bundle of "everyone else combined," envy-freeness limits the bundle of "every other individual," and CLB/CHB limit the bundle of "every group." Yet another intuition for CLB/CHB is that each agent wants the top $x$% of other agents not to have more than $y$% of the cake. (Also, to answer your last question, note that all three notions impose limits on what others receive.)
> >
> > I agree with this. However, this still does not really strike me as a proper motivation for CLB/CHB beyond the fact that they lie in-between proportionality and envy-freeness. Specifically, why would I (as an agent) are about how much the top $x$% of other agents receive? In particular, CHB (which is the fairness property for which most of the positive results are shown) still does not really seem to me to be particularly better motivated than proportionality. Yes, it always exists, and we can compute it efficiently which is interesting on its own. This, however, does not really mean that we want to compute it or that outcomes satisfying CHB are good (or a substantial improvement over PROP), and I think this needs to be better motivated for NeurIPS. (I hope this somewhat makes sense to you)
> >
> > > The "weird behavior" of Algorithm 1 you point out is interesting; however, one can raise similar concerns for perfect allocations ($v_i(X_j) = 1/n$), which are considered extremely fair.
> >
> > I agree with this, this seems more like a short-coming of "perfectness", which kind of makes perfectness seem more like a mathematical curiosity than a fairness notion you would want to achieve. I agree that Pareto optimality is probably too much to ask for (however, the illustrated efficiency violation seems to be much worse than just a Pareto optimality violation).

---

> > > ### Author Response · Authors · 2025-08-04
> > >
> > > Thank you again for your thoughtful comments.
> > >
> > > We note again that our primary interest in this work is theoretical. In fact, the initial inspiration for this work was that CHB would allow us to prove an exponential lower bound for the query complexity of finding EF allocations (which, of course, turned out not to be possible). That said, we do believe there’s a broader motivation as well: comparing one's allocation to that of the top fraction of other agents (e.g., the top 1%) is a familiar and intuitive way people discuss inequality in society. CHB and CLB formalize this kind of comparative reasoning.
> > >
> > > Finally, we agree with your comments regarding Pareto optimality/economic efficiency. This is an important issue, but somewhat outside the scope of this work.

---

### Official Review · Reviewer_CnGJ · 2025-06-20

**Clarity:** 4
**Significance:** 3
**Originality:** 3
**Rating:** 5
**Confidence:** 4

**Summary:**

This paper studies the cake-cutting problem, where a cake, modelled by [0, 1], is to be allocated to $n$ agents, each of whom admits an unknown distinct preference, and considers the popular RW query model. It was previously known that a proportional (PROP) allocation can be found in $O(n \log n)$ queries, an envy-free (EF) allocation can be found using $O(n^{n^{n^{n^{n^n}}}})$ queries, and a perfect allocation cannot be computed within a bounded number of queries.

This paper proposes new fairness notions that interpolate between PROP and EF/perfection, which are respectively named Complement Harmonically Bounded (CHB) and Complement Linearly Bounded (CLB). An allocation satisfies CHB-$k$ (resp. CLB-$k$) for some $k \in [n]$ if for all subsets $S$ with $|S| \leq k$ and $i \in S$, the value of $i$ for the union of pieces allocated to agents not in $S$ is at most $\frac{n - |S|}{n - |S| + 1}$ (resp. $\frac{n - |S|}{n}$), assuming each agent's total value for the cake is normalized to be $1$. CHB interpolates between PROP and EF with CHB-$1$ coinciding with PROP and CHB-$n$ being strictly weaker than EF. CLB interpolates between PROP and perfection with CLB-$1$ coinciding with PROP and CLB-$n$ being strictly weaker than perfection.

This paper gives an algorithm that computes a CHB-$n$ allocation in $O(n^4)$ queries and shows that no algorithm can compute a CLB-$2$ allocation with a bounded number of queries. Then, the paper considers an approximation version of CLB and provides an algorithm for finding a $\delta$-CLB-$n$ allocation using $O(\frac{n^6 \ln (1 / \delta)}{\delta \ln(n)})$ queries. Both algorithms exploit the observation that finding a complete, proportional, and nearly perfect allocation suffices, and invokes the algorithm for finding nearly perfect allocations given by prior work.

**Questions:**

- Line 272: Should $O(n^2)$ be $O(n^4)$? This is because there are $n$ iterations, and in each iteration, each agent takes $O(n^2)$ queries to mark. Or perhaps one can use the same trick as in the proof of Theorem 2 to reduce the last $O(n^2)$ cost to $O(1)$?

- In the statement of Theorem 2, how are $d$ and $\epsilon'$ defined? Can they be computed efficiently given the instance parameters?

- Line 355: Should $O(n^2)$ be $O(n^3)$? This is because there are $n$ iterations, and in each iteration, each agent needs to query $O(n)$ times.

- For the CHB-$k$ and CLB-$k$, do they imply any approximate version of EF? I'd especially like to know how far (whatever it means) CHB-$n$ is from EF in order to evaluate the significance of the $O(n^4)$ query complexity for CHB-$n$.

- Is there an intuitive description (ideally a description without using any math) of CHB-$k$ or CLB-$k$ for $k>1$?

**Ethical Concerns:**

["NO or VERY MINOR ethics concerns only"]

**Final Justification:**

My concerns are fully addressed, and I'd like to see the comments being incorporated in the future version of the paper.

**Limitations:**

Yes.

**Paper Formatting Concerns:**

No.

**Quality:**

4

**Strengths And Weaknesses:**

The study of cake-cutting has been initiated decades ago, and it still remains an important and interesting topic in algorithmic game theory. This paper contributes to this line of literature by proposing new fairness notions that interpolate between PROP and EF/perfection and providing a variety of intersting results. Personally, I find both notions quite natural in hindsight as they come from certain conditions implied by EF and perfection, and they nicely bridge between PROP and EF/perfection. On the other hand, it is less obvious that these notions can be interpreted as fairness even though they imply PROP, one of the most well-studied fairness concepts. This drawback makes it hard to judge whether such notions alone would find applications in either theory or practice.

The following algorithmic and hardness results in the paper are particularly good to know: (1) CHB-$n$ allocations, which is intuitively quite close to EF, can be computed in polynomially many queries, and (2) CLB-$2$ allocations, a seemingly mild relaxation of PROP toward perfection, inherits the hardness of computing perfect allocations. I believe these results will be of great interest to the fair division community as they seem to serve as an important step toward bridging the dichotomy between PROP and EF/perfection and closing the gap between upper and lower bounds for the query complexity of EF.

Moreover, the paper is in general well-written, and I really enjoy reading it, especially for the technical overview in the introduction. The proofs are correct to the extent that I checked. Overall, I believe that this paper exceeds the bar of NeurIPS and should be accepted.

Here are some detailed comments:
- Line 99-101: it is unclear why such a step forces us to deal with the weighted case.
- Line 126: It is unclear what "uniform" means here.
- Line 143: "$\delta$-CLB-$n$" -> "$\delta$-CLB-$k$".
- Line 144: "such that $k$" -> "such that if $k$".
- Line 161-164: The statement here seems obvious and redundant, since finding EF allocations has already been a strict subproblem of finding CHB-$n$ allocations.
- Line 13 in Algorithm 1: $S$ is a set of pieces instead of a piece, although it only contains one element at that time.
- In the proof of Theorem 1, it is not argued that Line 6 of Algorithm 1 can always be implemented, i.e., such a mark on $R$ is well-defined. Although this should follow easily from the same argument for the last remaining agent, it should be added for completeness. Same for the proof of Theorem 2.
- Line 325: "runtime" -> "query complexity".
- Line 333-334: This sentence seems to be grammatically problematic.
- Line 344: Should $I_{2,left}$ be $I'$?

The following papers are also relevant and are recommended to be cited:

A recent paper which also proposes a hierarchical type of notion for fair division, although it is for the strategic setting:
- It's Not All Black and White: Degree of Truthfulness for Risk-Avoiding Agents. Eden Hartman, Erel Segal-Halevi, Biaoshuai Tao.

Papers on computing approximate EF allocations:
- Envy-Free Cake-Cutting for Four Agents. Alexandros Hollender, Aviad Rubinstein.
- Hardness of Approximate Sperner and Applications to Envy-Free Cake Cutting. Ruiquan Gao, Mohammad Roghani, Aviad Rubinstein, Amin Saberi.
- Algorithmic Solutions for Envy-Free Cake Cutting. Xiaotie Deng, Qi Qi, Amin Saberi.

Papers on cake cutting with strategic agents:
- ​The Incentive Guarantees Behind Nash Welfare in Divisible Resources Allocation. Xiaohui Bei, Biaoshuai Tao, Jiajun Wu, Mingwei Yang.
- On Existence of Truthful Fair Cake Cutting Mechanisms. Xiaolin Bu, Jiaxin Song, Biaoshuai Tao.

---

> ### Author Rebuttal · Authors · 2025-07-28
>
> Thank you for your thorough and thoughtful review.
>
> Regarding your first question, great catch! You are correct in both comments: the same trick can be used to reduce the $O(n^2)$ to $O(1)$, but this is not explained properly, so it should be $O(n^4)$. We will update the paper to correct this.
>
> Regarding your second question, about $d$ and $\epsilon’$, in Appendix C we give a closed form for $d$ right above line 906, and a bound for $\epsilon’$ in line 909.
>
> Regarding your third question, about line 355, we are counting total queries, so we believe what is written is correct. Perhaps the confusion is that super-queries can be implemented using a constant number of actual queries; if so, see lines 335-345. Or, maybe the confusion is that we are tracking the evaluations of all intervals; if so, see lines 356-365.
>
> Regarding your fourth question, about the connection to approximate EF: a proportional allocation is, in the worst-case, $(n-2)/n - EF$, since, in the worst-case, $v_i(A_i) = 1/n$, but $v_i(A_j) = (n-1)/n$ for some other agent $j$. For a $CHB-n$ allocation, things can be as bad as $v_i(A_i) = 1/n$, $v_i(A_1) = ½$, and $v_i(A_j) = 1/(2n)$ for all remaining agents $j \neq 1$. This is a $(½ - 1/n)-EF$ allocation. We believe this is the worst case. This is a great observation that we had not made before, and we will include it in the revised version of the paper.
>
> Regarding your fifth question, about intuitive descriptions for these notions, observe that one can think of proportionality as a guarantee to every agent $i$ that, from $i$'s perspective, "the remaining $n-1$ agents, combined, don’t have too much of the cake" (as opposed to the standard phrasing/viewpoint: "agent $i$ has enough of the cake"). Envy-freeness sits at the other extreme: the guarantee to every agent $i$ is that, from $i$'s perspective, "for every other agent $j$, agent $j$ doesn't have too much of the cake." CHB interpolates between the two extremes: the guarantee to every agent $i$ is that, from $i$'s perspective, "for every group of agents, the group, combined, doesn’t have too much of the cake." Put differently, proportionality limits the bundle of "everyone else combined," envy-freeness limits the bundle of "every other individual," and CLB/CHB limit the bundle of "every group." Yet another intuition for CLB/CHB is that each agent wants the top $x$% of other agents not to have more than $y$% of the cake.
>
> We will also address the minor comments you point out (citations, typos, etc) in the final version of the paper.

---

> > ### Comment · Reviewer_CnGJ · 2025-08-02
> >
> > I thank the authors for your response. My questions are sufficiently addressed, and I will maintain my score.

---

### Official Review · Reviewer_oPsd · 2025-06-28

**Clarity:** 3
**Significance:** 3
**Originality:** 3
**Rating:** 4
**Confidence:** 4

**Summary:**

This paper introduces two new fairness notions, including Complement Harmonically Bounded-$k$ (CHB-$k$) and Complement Linearly Bounded-$k$ (CLB-$k$), to the cake-cutting model. There are several contributions in this paper.

(1) They study the relationship between the above two notions and the existing fairness concepts, including EF and PROP.

(2) For CHB-$n$, they design an algorithm to compute such an allocation in $O(n^4)$ queries with RW model and provide a lower bound of $\Omega(n^2)$ queries for CHB-2 allocations.

(3) For CLB-$k$, they construct an instance where CLB-2 allocations cannot be computed in bounded queries. Then, they focus on the approximate CLB-$n$ allocation and propose an algorithm to compute it in bounded queries.

**Questions:**

Q1: Could you please give me some practical examples or non-mathematical explanation of the proposed notions?


Q2: For the computation of approximate CLB-$n$ allocations, I am a little curious about the lower bound of queries. So, could you please provide some insights for it?

**Ethical Concerns:**

["NO or VERY MINOR ethics concerns only"]

**Final Justification:**

The research problem is interesting, and the major concern for me is the motivation, since EF or PROP has strong theoretical and real application motivation, but the proposed notions may not be motivated sufficiently. Considering that the authors will strengthen the motivation, I will keep my score.

**Quality:**

3

**Strengths And Weaknesses:**

Strength:

(1) This paper provides a new perspective of fairness besides EF and PROP, which fills the gap in the cake cutting model.

(2) They smartly adapt the current technique to develop new algorithms to compute CHB-$n$ and the approximate CLB-$n$ allocations.

(3) The whole paper is well written, and it is easy for us to read.

Weakness:

(1) In the relationship graph, we can easily find the position of CHB and CLB, which gives us a mathematical view of them. It may be a little bit hard to understand the practical motivation or meanings of these two notions, compared to the classic notions, including EF and PROP. For example, for PROP, each agent has a value threshold, and then compares the value of her bundle with it, which is easy for us to understand.

(2)  For the CHB-$n$ allocations, the algorithm design is nice, but there is still a gap between the lower and upper bounds of queries. For the approximate CLB-$n$ allocations, it seems that the lower bound of queries is missing.

---

> ### Author Rebuttal · Authors · 2025-07-28
>
> Thank you for your thorough and thoughtful review.
>
> Regarding your first question about practical examples/non-mathematical explanations for CHB/CLB: one can think of proportionality as a guarantee to every agent i that, from $i$'s perspective, "the remaining $n-1$ agents, combined, don’t have too much of the cake" (as opposed to the standard phrasing/viewpoint: "agent $i$ has enough of the cake"). Envy-freeness sits at the other extreme: the guarantee to every agent $i$ is that, from $i$'s perspective, "for every other agent $j$, agent $j$ doesn't have too much of the cake." CHB interpolates between the two extremes: the guarantee to every agent $i$ is that, from $i$'s perspective, "for every group of agents, the group, combined, doesn’t have too much of the cake." Put differently, proportionality limits the bundle of "everyone else combined," envy-freeness limits the bundle of "every other individual," and CLB/CHB limit the bundle of "every group." Yet another intuition for CLB/CHB is that each agent wants the top $x$% of other agents not to have more than $y$% of the cake. Of course, different definitions of "too much" (linear vs harmonic) lead to notions that are not strictly between PROP and EF, but we hope that this perspective/viewpoint gives some more intuition and motivation for our new notions.
>
> Regarding your second question about lower bounds for approximate CLB-allocations: this is a great question! Our work does not provide any lower bounds, and we view this as a very interesting direction for future work. That said, we note that even for (the arguably more fundamental notions of) approximate proportionality and approximate envy-freeness, the literature does not provide any answers. To the best of our knowledge, the only known lower bound for approximate envy-freeness is by [BN22], which prove a $\Omega(log(1/\epsilon))$ bound for finding an epsilon EF allocation with *contiguous* pieces. That would imply a lower bound for finding contiguous $\delta-CLB-(n/2)$ allocations; given the restrictive setting and mild lower bound, we chose not to emphasize this point in the paper.
>
> We would be happy to incorporate these clarifications in the final version of the paper.

---

> > ### Comment · Reviewer_oPsd · 2025-08-04
> >
> > Thanks for your response! I hope you can further strengthen the motivation of the definition and polish the whole paper. I will maintain my score.

---

### Official Review · Reviewer_3btH · 2025-07-04

**Clarity:** 4
**Significance:** 3
**Originality:** 4
**Rating:** 5
**Confidence:** 4

**Summary:**

This paper studies the problem of fair cake-cutting. There are some well-studied notions of fairness in this problem including proportionality and envy-freeness. This paper introduces two hierarchical fairness notions to bridge the gaps in the number of queries  (in the RW model) needed to achieve different notions. These notions are called Complement Harmonically Bounded-k (CHB-$k$) and Complement Linearly Bounded-$k$ (CLB-$k$). An allocation is CHB-$k$ if for any subset of size $s \le k$ of agents and agent $i$ in this subset, value of $i$ for the union of the pieces received by the agents not in this set is at most $\frac{n-s}{n-s+1}$. This bound changes to $\frac{n-s}{n}$for CLB-$k$. Both CLB-1 and CHB-1 are equal to proportionality and CLB-$n$ coincides with super envy-freeness.

They show upper and lower bounds on the number of queries required for these notions, and also study approximate versions of them. Figure 1 show a great overview of the main results.

**Questions:**

Do you have some intuition on how relaxed versions of these notions look in the indivisible setting? Let's say we want to talk about CLB-$k$ up to one item/any item.

**Ethical Concerns:**

["NO or VERY MINOR ethics concerns only"]

**Final Justification:**

I like the paper in general. I'm still not sure if this is the right venue for this but I rather let the authors choose their audience. As other reviewers have mentioned some definitions lack enough motivation. I can see the paper going either way but I would vote for acceptance.

**Limitations:**

-

**Paper Formatting Concerns:**

-

**Quality:**

4

**Strengths And Weaknesses:**

## Strengths

The paper is well-written and easy to follow.

The idea of introducing such fairness notions is novel and interesting and the results advance our understanding of fair cake cutting in the Robertson-Webb model.

The algorithms are mostly intuitive and proofs are solid.

The paper provides a nice overview of the related literature.


## Weaknesses

I pretty much like the contribution of this paper. My only concern is whether this is the right venue for this work. The paper is rather technical and due to lack of space most of the interesting part had to move to the appendix. In general I would suggest WINE/EC/SODA for this paper.

I think after the definition of these notions, lower-bounds are the most interesting part of this work. I was a bit disappointed that they were all in the appendix.

In the definitions of CLB and CHB, $S$ is defined as the set that includes $I$ but then n-$S$ is used in the bounds. I don't see the reason for this as you can instead define $S$ as a set that does not include $i$  and use $|S|$ instead of $n-|S|$ everywhere.

---

> ### Author Rebuttal · Authors · 2025-07-28
>
> Thank you for your thorough and thoughtful review.
>
> Regarding your question, the short answer is that we have not explored this direction in depth. We believe it’s an interesting direction for future work. Here are some preliminary thoughts: looking at the proof that CHB is implied by EF, if one were to replace “EF” with “EF-1” or “EF-X,” the approximation would blow up (so, e.g., “CHB up to |S| items”). At the same time, “CHB up to one item” seems easy to achieve, intuitively, so perhaps simply looking at the proof of existence for CHB-k from EF-k is not the right approach. The existence of “CHB up to *any* item” seems like the first interesting question here. We are not sure whether the answer is more similar to the case of PROPX (known not to exist, even in simple settings) or EFX (known to exist for a small number of agents and certain valuation classes, but generally difficult to resolve).
>
> Regarding your comment about the definitions of CLB/CHB, perhaps our responses to Reviewers oPsd and JByF help clarify our motivation and intuition for these notions.

---

> > ### Comment · Reviewer_3btH · 2025-08-04
> >
> > Thanks for the response. I'll maintain my score.

---

### Decision · Program_Chairs · 2025-09-17

**Decision:**

Accept (poster)

**Comment:**

The paper proposed two new hierarchies of notions of fairness in cake cutting that interpolate well-studied notions, and investigated their properties. Reviewers agreed that the problem studied in this paper is significant and acknowledged the novelty and clarity of the paper. The main reservation is the motivation behind the two new notions of the fairness. Unfortunately the issue was not addressed after the rebuttal and discussions. Multiple reviewers felt that the paper might be a better fit for a more theoretical/specialized venue such as EC, WINE, or SODA. Still, the paper can be a good fit for NeurIPS and would stimulate more discussions and future work.

If the paper is accepted, the authors are strongly encouraged to think about justifications in camera ready and for future work. Proposing new notions of fairness and investigating their relationship/properties are great research topics, but motivating the new notions by real-life scenarios will have greater impact and do a better service to research community and our society.